# Influence of LPBF-Surface Characteristics on Fatigue Properties of Scalmalloy®

**Jens Musekamp** [1,*] , **Thorsten Reiber** [2] , **Holger Claus Hoche** [1] , **Matthias Oechsner** [1] , **Matthias Weigold** [2] and **Eberhard Abele** [2]

1    Center for Engineering Materials, State Materials Testing Institute Darmstadt (MPA) Chair, Institute für Materials Technology (IfW), Technical University of Darmstadt, Grafenstraße 2, 64293 Darmstadt, Germany; holger_claus.hoche@tu-darmstadt.de (H.C.H.); matthias.oechsner@tu-darmstadt.de (M.O.)

2    Institute of Production Management, Technology and Machine Tools (PTW), Technical University of Darmstadt, Otto-Berndt-Straße 2, 64287 Darmstadt, Germany; t.reiber@ptw.tu-darmstadt.de (T.R.); weigold@ptw.tu-darmstadt.de (M.W.); abele@ptw.tu-darmstadt.de (E.A.)

*    Correspondence: jens.musekamp@tu-darmstadt.de; Tel.: +49-6151-16-24890

**Abstract:** Laser powder bed fusion (LPBF) has indisputable advantages when designing new components with complex geometries due to toolless manufacturing and the ability to manufacture components with undercuts. However, fatigue properties rely heavily on the surface condition. In this work, in-process surface parameters (three differing contour parameter sets) and post-process surface treatments, namely turning and shot peening, are varied to investigate the influence of each treatment on the resulting fatigue properties of LPBF-manufactured specimens of the aluminium–magnesium–scandium alloy Scalmalloy®. Therefore, metallographic analysis and surface roughness measurements, as well as residual stress measurements, computer tomography measurements, SEM-analyses, tensile and fatigue tests, along with fracture surface analysis, were performed. Despite the fact that newly developed in-process contour parameters are able to reduce the surface roughness significantly, only a minor improvement in fatigue properties could be observed: Crack initiation is caused by sharp, microscopic notches at the surface in combination with high tensile residual stresses at the surface, which are present on all in-process contour parameter specimens. Specimens using contour parameters with high line energy show keyhole pores localized in the subsurface area, which have no effect on crack initiation. Contours with low line energy have a slightly positive effect on fatigue strength because less pores can be found at the surface and subsurface area, which even more greatly promotes an early crack initiation. The post-process parameter sets, turning and shot peening, both improve fatigue behaviour significantly: Turned specimens show lowest surface roughness, while, for shot peened specimens, the tensile residual stresses of the surface radially shifted from the surface towards the centre of the specimens, which counteracts the crack initiation at the surface.

**Keywords:** fatigue; aluminium alloys; stress measurements; post processing; laser powder bed fusion

## 1. Introduction

Laser powder bed fusion (LPBF) belongs to the group of additive manufacturing (AM) processes in which a three-dimensional component is produced by joining volume elements and building up thin surfaces layer by layer [1]. By manufacturing with LPBF, complex geometries such as thin walls or internal grid structures which have an almost 100% density can be realized, allowing a component to be optimized both in terms of weight and function [1–3]. This new freedom in design benefits a wide range of the manufacturing sector to improve and develop new products e.g., optimized valves and pumps [4], new tools, and optimized turbine blades or automotive applications [5], respectively. Especially for aluminium alloys, there is a high need for sustainability, efficiency increase, and cost reduction in the automotive, aerospace, and aircraft industries [6]. Moreover, there is a

need to find alternatives for titanium alloys in terms of the strength–width ratio [7], which contributes much, along with the design rules of LPBF.

To fulfil the requirements of high strength aluminium alloys, Scalmalloy® was developed especially for manufacturing with the LPBF-process [8]. The aluminium–magnesium–scandium alloy has high tensile strength along with high elongation at break compared to other high strength aluminium alloys [9]. $Al_3(Sc_xZr_{1-x})$ and AlMgO phases act as effective nucleating agents during solidification of the melt [10] and are responsible for the high mechanical properties obtained with additional precipitation of these phases during precipitation hardening heat treatment (325 °C, 4 h) [11–15]. The infill parameters were optimized with regard to the density to achieve up to 99.8% dense specimens [16–19].

Apart from a high density, the surface zone, which has high roughness in the LPBF-manufactured state [20], and in which an increased pore density may be present [21,22], exerts a significant influence on the fatigue behaviour [20]. The roughness is caused by three characteristic effects: the irregular geometric solidification of the melt pool due to the process dynamics [23,24], powder particles adhering to the surface [23,25], and the staircase effect [26,27]. The latter only comes into play with inclined surfaces. On the one hand, irregular melt tracks occur due to the balling effect at low line energy $E_L$, which is determined by the ratio of laser power P and scanning speed v. This effect is caused by the minimization of the surface energy of the melt, which is driven by its surface tension [28,29]. On the other hand, irregular melt tracks can result in an increased line energy owing to the highly dynamic Marangoni convection [30]. It must be taken into account when only using an infill parameter set that the outline of vertical and inclined surfaces is defined by the melt track fronts of numerous individual melt tracks depending on the layer rotation angle. Roughness values for vertical walls in the case of aluminium alloys are at $8\ \mu m \le R_a \le 25\ \mu m$ [31–33].

Murakami et al. [34] states that on AM-materials in general, the crack initiation more often starts at the surface due to surface roughness or defects at the surface, than at internal pores.

The high roughness and defects in the surface zone are considered to act as notches and thus contribute to the reduction and scattering of the fatigue strength [20]. Therefore, the surface zone of safety-relevant components is usually removed by a turning post-processing [20], which results in a considerable expense of time and costs. Wagener et al. [35] showed that on LPBF-AlSi10Mg especially, the near-surface porosity, which is brought in by contour scans in overhanging regions, lowers the fatigue properties significantly. Moreover, notches, which are located near the surface (subsurface porosity) exhibit a high influence on the fatigue properties.

The influence of pores on fatigue properties can be simulated using the $\sqrt{\text{area}}$-concept of Murakami [36]. In [37], a comparison of fatigue properties of forged, castes, wrought and LPBF-manufactured material regarding the fatigue properties was summarized and simulated with the $\sqrt{\text{area}}$-concept. The large scatter of the fatigue properties for machined materials could be significantly reduced if the data are correlated to the surface of the defect size at the failure origin. The fatigue strength for LPBF-manufactured AlSi10Mg and LPBF-manufactured Ti6Al4V-specimens can even exceed the fatigue strength of conventional manufactured specimens if they are machined and stress-relieved. Subsurface porosity, which can result due to a poor contour scan, can reduce the fatigue properties significantly. Moreover, for additively manufactured 316L, the fatigue properties are somewhat equivalent to those of wrought materials [38].

Maskery et al. [39] investigated the pore size distribution of LPBF-AlSi10Mg and found out that it follows a Weibull distribution, which can be used for lifecycle-modelling, determining the crack growth rates.

In [20,40], it was stated that subsurface porosity and surface roughness decrease the fatigue properties of AlSi10Mg significantly. Different post-process treatments can reduce these lowering factors: a vibratory finishing increases the fatigue properties due to lowering the surface roughness. Sand blasting increases the fatigue properties even more due to the

induction of compressive residual stresses, although the surface roughness is higher. The highest increase in fatigue strength was achieved by machining and polishing.

The influence of machining on fatigue properties of LPBF-AlSi10Mg was shown in [41], where machining significantly increases the fatigue limit. This was mainly explained with the decrease in surface roughness, because satellites (powder at the surface) and balling (agglomerates of molten material) can be identified on as-built specimens, which could act as crack initiation areas. After machining, some porosity is still present at the surface, which can cause crack initiation.

Brandão [42] improved the fatigue properties of AlSi10Mg by jet blasting after manufacturing, which increases the fatigue properties in the same way as a vibratory finishing.

In [16], the fatigue properties of heat treated and turned Scalmalloy® (laser power = 370 W, hatch distance = 100 µm, 30 µm layer thickness and 1600 mm/s scanning speed) have been investigated at a stress ratio of R = 0.1 up to N = $3 \times 10^7$ cycles. The fatigue properties show high scatter and a significant correlation with the building orientation: Specimens in building direction with a hot isostatic pressing (HIP)-treatment exhibit the lowest fatigue limit of $\sigma_{max}$ = 160 MPa ($\sigma_a$ = 88 MPa), whereas specimens in a diagonal direction show a fatigue limit up to $\sigma_{max}$ = 240 MPa ($\sigma_a$ = 132 MPa). The crack initiation of Scalmalloy® was dependent on local microstructural features around the crack tip. The main driver for the poor fatigue crack growth behaviour is explained with the fully coherent $Al_3(Sc_{(1-x)}Zr_x)$ precipitations: In Scalmalloy®, neither grain boundaries nor precipitations are able to prevent crack growth propagation under cyclic loads.

Compared to the post-processing methods described above, in-process methods such as contour scans can be performed directly during the LPBF-process in order to save additional time and costs. When using contour scans, the material is melted in one or more vectors parallel to the contour of the component. Exposure of the contour can take place before (pre-contour) or after the infill (post-contour). The process parameters of the contour scans can be optimized independently of the infill parameters in terms of low roughness. To achieve a low roughness, increased energy input can be applied. This leads to more uniform melt tracks and the peaks and valleys between the layers can be decreased by re-melting [26]. Using two contour scans with a high line energy $E_L$ = 1.17 J/mm, a roughness of Ra = 6.1 µm on vertical walls in Scalmalloy® specimens could be achieved. However, some pores were still present despite the contour scans [9]. The reason for the occurrence of pores when exposing using high line energies can be the transition from conduction-mode to keyhole-mode welding [43], which is described in detail in [44]. In order to avoid the formation of keyhole pores, welding in keyhole mode can be prevented by reduction of the line energy.

In this work, the influence of surface properties such as roughness and residual stress, the intrinsic pores, and the subsurface porosity on the fatigue properties for the LPBF-manufactured Al–Mg–Sc alloy Scalmalloy® are investigated to extend the knowledge of these influencing factors. This helps to optimise the fatigue strength when designing for subsequent operational use. Therefore, three contour scanning strategies derived from a preliminary study of the authors [44] are compared to the as-built surface state without contour parameters, the shot peened-state, and the turned-state. The properties are examined using roughness measurements, residual stress measurements, optical Microscopy (OM), fracture surface analyses by means of Scanning Electron Microscopy (SEM), tensile tests, and fatigue tests.

## 2. Materials and Methods

Commercially available Scalmalloy® powder material from Heraeus Additive Manufacturing GmbH with a mean particle size of 20 to 65 µm (D10 to D90) was used. The chemical composition of an exemplary additively manufactured specimen measured by spark-emission spectroscopy is shown in Table 1.

**Table 1.** Composition of the manufactured specimens measured via spark-emission spectroscopy along with the nominal values of Scalmalloy®.

| - | | Si | Fe | Mn | Mg | Zn | Ti | V | Sc | Al |
|---|---|---|---|---|---|---|---|---|---|---|
| Nominal | Min. | 0 | 0 | 0.3 | 4 | 0 | 0 | 0 | 0.6 | |
| | Max. | 0.4 | 0.4 | 0.8 | 4.9 | 0.25 | 0.15 | 0.05 | 0.8 | Cal. |
| | | data | data | data | | | | | | |
| Measured | | $0.04 \pm 0.01$ | $0.1 \pm 0.004$ | $0.5 \pm 0.003$ | $3.95 \pm 0.05$ | <0.01 | $0.08 \pm 0.003$ | $0.02 \pm 0.001$ | $0.65 \pm 0.05$ | |

All investigated specimens were produced on an LPBF system EOS M 290 using an Yb-fibre laser, having a recommended maximum laser power of 370 W and a focus diameter of 100 μm. The maximum build volume of the system is 250 mm × 250 mm × 325 mm. In order to keep the oxygen content below 0.1% during the building process, the building chamber is inerted with argon gas and a laminar argon flow is utilized.

The process strategy time homogenization is a function invented by EOS GmbH and implemented in the EOSPRINT 2 software [45]. The volume energy $E_V$ is determined by the ratio of laser power P and the product of scanning speed v, hatch distance h and layer height $l_z$. For fatigue testing, six different surface states were considered, whereas four states were directly created during the manufacturing without any post-processing. The as-built state of the surface only consists of the infill parameter set (Table 2). The three different specimens that have contour parameter sets were manufactured with the infill parameter set and subsequently the contour parameters, designated as contour A, B, and C (Table 3). The latter represent potential parameter sets developed in a preliminary study by the authors applying DoE with the objective to reduce the roughness of vertical walls without introducing additional pores into the contour area [44]. The factors with the largest influence on the surface roughness and density were identified as the scan speed v, the laser power P, and the percentage overlap of the contour scan with the infill $o_{ci}$ (Figure 1).

**Table 2.** Used infill parameter sets for Scalmalloy®, applying the strategy time homogenization.

| | |
|---|---|
| Volume energy, $E_V$ [J/mm³] | 110 |
| Line energy, $E_L$ [J/mm] | 0.23 |
| Laser power, P [W] | 370 |
| Scan speed, v [mm/s] | 1600 |
| Hatch distance, h [mm] | 0.07 |
| Layer height, $l_z$ [mm] | 0.03 |
| Exposure pattern | Stripes |
| Stripe width [mm] | 7 |
| Layer rotation | 67° |

**Table 3.** In-process contour parameter sets with different line energy $E_L$, laser power P, scan speed v, and percentage overlap of the contour scan with the infill $o_{ci}$ used in this study [44].

| - | Type | $E_L$ [J/mm] | P [W] | v [mm/s] | $o_{ci}$ [%] |
|---|---|---|---|---|---|
| Contour A | Post-cont. | 0.9 | 370 | 411 | 50 |
| Contour B | Pre-cont. | 0.6 | 370 | 600 | 50 |
| Contour C | Pre-cont. | 0.07 | 230 | 3286 | 50 |

The results of the development of the contour parameters of the former study are summarized as follows [44]:

- Contour A: the lowest overall roughness of $R_a < 7$ μm ($R_z < 55$ μm) can be attained by applying a post-contour with a high line energy in the range of $E_L = 0.9$ J/mm. The high line energy has a negative effect on the formation of keyhole pores, which start to form in the $E_L$-range between 0.6 J/mm and 0.75 J/mm. For contour A, a second contour scan with a lower line energy of $E_L = 0.55$ J/mm² was applied with an offset to the first contour of 120%, which means a shift towards the inside of the sample.

- Contour B: the lowest roughness of $R_a < 9.8$ μm ($R_z = 65.2$ μm) utilizing a contour parameter set with no significant increase in contour porosity is achieved when using a pre-contour with $E_L = 0.6$ J/mm and $o_{ci} = 50\%$. Isolated larger pores can be detected in the area of the vertical specimen edge.
- Contour C: A reduction of roughness in the range of low line energies $E_L < 0.14$ J/mm can be achieved by using a pre-contour. The lowest roughness $R_a = 12.3$ μm ($R_z = 78.4$ μm) is obtained at $E_L = 0.07$ J/mm in combination with $o_{ci} = 50\%$.

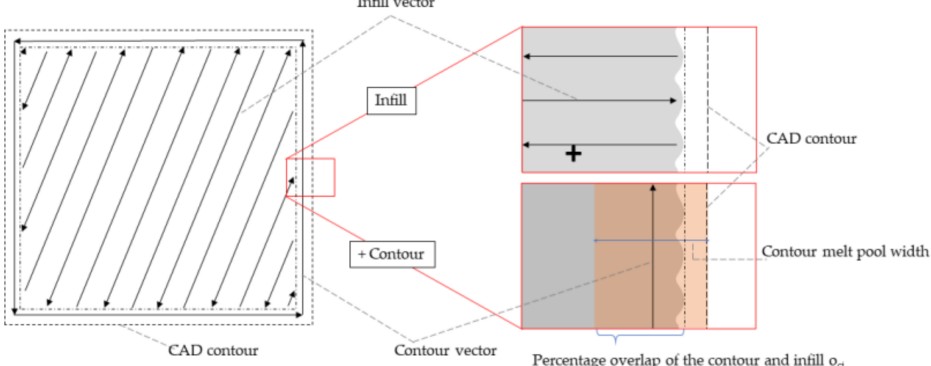

**Figure 1.** Simplified illustration of the contour and infill vectors based on [44]. Additionally, the percentage overlap of the contour scan and the infill $o_{ci}$ depending on the contour melt pool width is illustrated.

The individual parameters of the three contour parameter sets applied in this study are summarized in Table 3:

The entire building platform with the fabricated specimens was subjected to heat treatment immediately after the building process at 325 °C for 4 h with slow cooling to set maximum quasistatic properties [46]. Table 4 gives an overview about all manufactured fatigue specimens.

**Table 4.** Overview of all manufactured and tested fatigue specimens.

| Name | Surface | Heat Treatment | Building Direction |
|------|---------|----------------|--------------------|
| AB | As-built | | |
| T | Turned | | |
| SP | Shot peened | 325 °C, 4 h | Z |
| A | Contour A | | |
| B | Contour B | | |
| C | Contour C | | |

Figure 2 shows the geometry of the manufactured fatigue specimens. All specimens were manufactured with identical radii in the test areas.

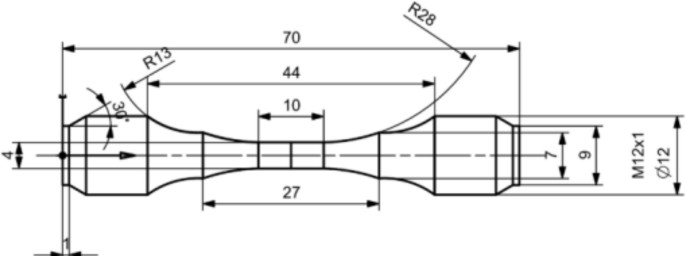

**Figure 2.** Manufactured and tested fatigue specimen used in this study. The testing area was applied with the different surface.

As-built specimens and specimens with contour parameters were manufactured in the final geometry (Figure 2). Only the thread of the fixation to clamp the specimen into the machine was turned afterwards. To avoid crack formation in the overhang region adjacent to the examination area, manual post-processing was carried out by applying rotating tools for grinding and rubbering on both sides adjacent to the examination area. The two post-processing states represent the turned and the shot peened states. Turned specimens were manufactured as a cylinder with a 10 mm diameter and turned to the final geometry by reducing the diameter in the testing area by 6 mm. Shot peened specimens were manufactured as as-built specimens, and shot peened after the heat treatment using a peening media by Iepco AG Iepconorm A (nutshells, particle size = 200 μm–400 μm) followed by Iepconorm B-4 (C, Si, Mn, P, S, particle size = 125 μm–250 μm) and Iepconorm C ($ZrO_2$, $SiO_2$, $Al_2O_3$, particle size = 400 μm) at a pressure of 2 bar, respectively. The blasting process was carried out for 20 s per specimen using peening media and a working distance of 5 cm. The exact diameter of the test area of the specimens was measured before performing the tests.

In order to obtain the quasistatic properties and to calculate the X-ray elastic constant of residual stress measurements, standing (Z) and lying (X) cylinders were manufactured on the same building platform in addition to the fatigue specimens to manufacture tensile specimens with geometry B6 × 30 according to DIN 50125:2016 [47] for quasistatic investigations. Half of the cylinders for the tensile tests were cut from the platform before heat treatment. Afterwards, all cylinders were turned into the final geometry and the stress–strain curve was obtained with a universal testing machine H&P 250, using strain control according to DIN EN ISO 6892:2009-01 [48].

The roughness of the fatigue specimens in the test area was determined with a Hommel-Etamic Turbo Wave V7.59 on a length of 15 mm with three measurements per specimen variation. Because the test length was too short for a roughness measurement according to DIN EN ISO 4288:1998 [49], the measurement direction was reversed and the specimens were rotated at a certain angle.

The residual stresses were measured with x-ray-diffraction (XRD) using Cr-Kα-radiation at the {311}-reflex of Aluminium according to DIN EN 15305:2009-01 [50], using the $\sin^2\Psi$-method. Therefore, a diffractometer by Stresstech, model Xstress G3R in a modified $\Psi$ –arrangement, was used. The residual stresses were calculated from seven equidistant $\Psi$-angles from $0°$ to $\pm 45°$. The X-ray elastic constant used to calculate the residual stress values was $\frac{1}{2}S_2 = 1.9 \times 10^{-5}$ mm$^2$/N, which was calculated from the tensile tests and [51,52]. Initially, the residual stresses were measured in loading direction on three positions in the testing area of the fatigue specimens. The value was then evaluated with a cross correlation to gather the measurement uncertainty.

Cross sections for microstructural investigations were prepared to investigate the microstructure by optical microscopy (OM) using standard metallographic polishing techniques with a final polishing step and 1 μm OPS-suspension. OM-specimens were etched with TIM3-etchtant (water, HCl, HF) and analysed using a LEICA DM4000 microscope. SEM-analyses were performed with a Zeiss EVO 60 scanning electron microscope.

All manufactured specimens were examined regarding the porosity by 3D-computed tomography using a GE v|tome|x 240 d with a nanofocus-tube. Only the testing area in the centre of the specimens was considered in order to optimize the maximum possible resolution. The achieved voxel-size is 16 μm. To reduce beam hardening, an Sn-filter with 0.1 mm thickness was used. After reconstruction, the porosity was analysed with the vgdefx-algorithm of the porosity module of VGStudioMAX 3.0 to determine the properties and distribution of the pores. In Figure 3, a reconstructed model of the testing area with the pore distribution and a coloured pore size scale is shown exemplarily.

The fatigue tests were performed with a Zwick 100 HFP 5100 following the bead string method to draw the Woehler curves as described in DIN 50,100 [53]. The specimens have a notch factor of k = 1 and were tested with a load ratio of R = 0.1. Fracture surfaces were analysed by SEM to gather knowledge about crack initiation and propagation.

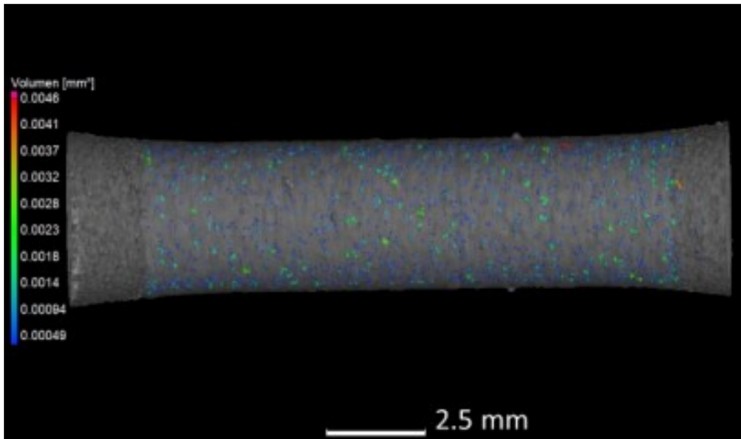

**Figure 3.** Exemplary reconstructed model of the testing area of an A-specimen with pore distribution and coloured pore sizes.

## 3. Results

### 3.1. Quasistatic Tests

Table 5 shows the results of the tensile tests according to DIN EN ISO 6892-1:2009 [48]. It is proven that the tensile strength $R_m$ and the yield strength $R_{p0.2}$ were significantly increased by the heat treatment, but the elongation at fracture A was decreased to the same extent, which corresponds to the results in [46]. The modulus of elasticity did not change significantly. The increase in yield strength is due to precipitation hardening of the $Al_3Sc_xZr_{1-x}$-precipitations that occur during the heat treatment [54]. It can also be observed that standing (Z)-specimens drop slightly in strength compared to the lying (X)-specimens, regardless of the heat treatment. This has also been observed in other LPBF-materials and can be explained by the Schmid factor, i.e., the orientation of the maximum acting shear stress to the sliding system [55]. For face-centred cubic materials such as aluminium, the {111} <110> plane is the sliding system that has the lowest strength: Because there is a preferred orientation in additively manufactured materials, a larger Schmid factor $\mu$ can occur in the standing (Z)-specimens compared to the building directions, which macroscopically manifests itself in a lower tensile strength and yield strength [52].

**Table 5.** Quasistatic properties of turned tensile specimens with geometry B6 × 30.

| Orientation | Heat Treatment | $R_m$ [MPa] | $R_{p0,2}$ [MPa] | A5 [%] | E [GPa] |
|:---:|:---:|:---:|:---:|:---:|:---:|
| X | As-built | 352 ± 1 | 290 ± 3 | 24.5 ± 1 | 70 ± 2 |
| Z | As-built | 349 ± 2 | 269 ± 1 | 22 ± 2 | 70 ± 1 |
| X | 325 °C, 4 h | 520 ± 2 | 490 ± 2 | 11.5 ± 1 | 71 ± 1 |
| Z | 325 °C, 4 h | 510 ± 2 | 476 ± 2 | 14 ± 1 | 71 ± 1 |

Compared to other AM-materials such as Inconel 718, the fluctuations in the mechanical properties depending on the building direction of less than 1% are practically negligible.

Figure 4a shows an exemplary overview of the fracture surface of a heat treated tensile specimen investigated via SEM. In detailed view, the dimples can be observed in the fracture microstructure, which is characteristic for a transcrystalline forced fracture. The dimples are less (Figure 4b) or more (Figure 4c) distinct depending on the area observed. These findings are a characteristic behaviour for ductile materials, such as aluminium alloys, according to VDI 3822 Part 2 [56].

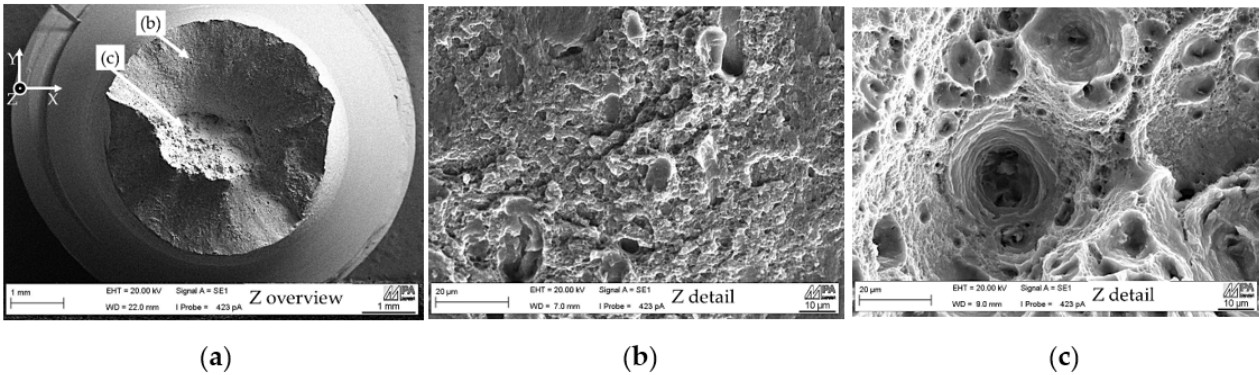

**Figure 4.** (**a**) Overview of an exemplary fracture surface of a standing (Z) and heat treated tensile specimen. (**b**) Exemplary detailed view with less dimples. (**c**) Exemplary detailed view with dimples.

### 3.2. Metallographic and Porosity Analysis

In Table 6 the surface roughness as well as the measured overall porosity is shown. It can be stated that AB has the highest surface roughness and T the lowest. All specimens with post-processing parameters T and SP as well as AB do not differ significantly in their overall porosity, because they were manufactured only with the infill parameter set. Figure 5 provides overviews of the cross-sections of the test area from each fatigue specimen examined by OM. The overall porosities in the infill of the specimens do not differ significantly. The high surface roughness due to adhering powder particles and agglomerations at the surface of the as-built specimen without contour scan (Figure 5a) is clearly visible. The shot peening process (Figure 5b) removes the powder particles and agglomerations to a high extent resulting in a smoother surface. The highest reduction of the surface roughness was achieved by turning, as can be obtained from Table 6 and Figure 5c.

**Table 6.** Results of the roughness measurements and overall porosity measurement via CT.

| Name | $R_a$ [μm] | $R_z$ [μm] | Overall Porosity (CT) [%] | Largest Pore [mm³] | Distance of Largest Pore to Surface [mm] |
|------|-----------|-----------|---------------------------|--------------------|------------------------------------------|
| AB | 17 ± 2 | 101 ± 7 | 0.007 ± 0.004 | 0.0012 | 0.174 |
| T | 1 ± 0.2 | 5 ± 0,8 | 0.005 ± 0.003 | 0.0007 | 0.160 |
| SP | 8 ± 2 | 43 ± 11 | 0.003 ± 0.001 | 0.00151 | 0.164 |
| A | 5 ± 2.5 | 23 ± 11 | 0.252 ± 0.05 | 0.00302 | 0.112 |
| B | 6 ± 0.3 | 30 ± 7 | 0.019 ± 0.001 | 0.00133 | 0.096 |
| C | 11 ± 0.1 | 63 ± 4 | 0.012 ± 0.003 | 0.00184 | 1.498 |

In general, the specimens built with contour parameters result in a smoother surface in comparison to AB (Table 6). Among all contour parameters, specimen A exhibits the highest overall porosity. In the etched state in Figure 5d, on the right hand side, it can be noticed that the large pores are located in the contour area. Contour parameter C leads to the highest surface roughness (Table 6 and comparison of Figure 5d–f). It is noticeable that in the etched state of specimens with contour parameter set A and B, microstructural effects due to the contour scans are clearly visible. For specimens with contour parameter set A, the smoother surface can be explained by the performed post-contour scan, which results in the characteristic contour ring near the surface (Figure 5d, right hand side). Specimens with contour parameter set B exhibit the same visible contour parameter ring in the cross section in Figure 5e (right hand side), which is slightly thinner than in A. In contour parameter C, the line energy of the pre-contour was significantly lower than the line energy of the infill parameter: The overlap of $o_{ci}$ = 50% (Table 3 in comparison to Table 2) was enough to completely re-melt the surface area, resulting in no defined contour parameter ring in the etched state of cross sections on specimens with contour parameter set C (Figure 5f).

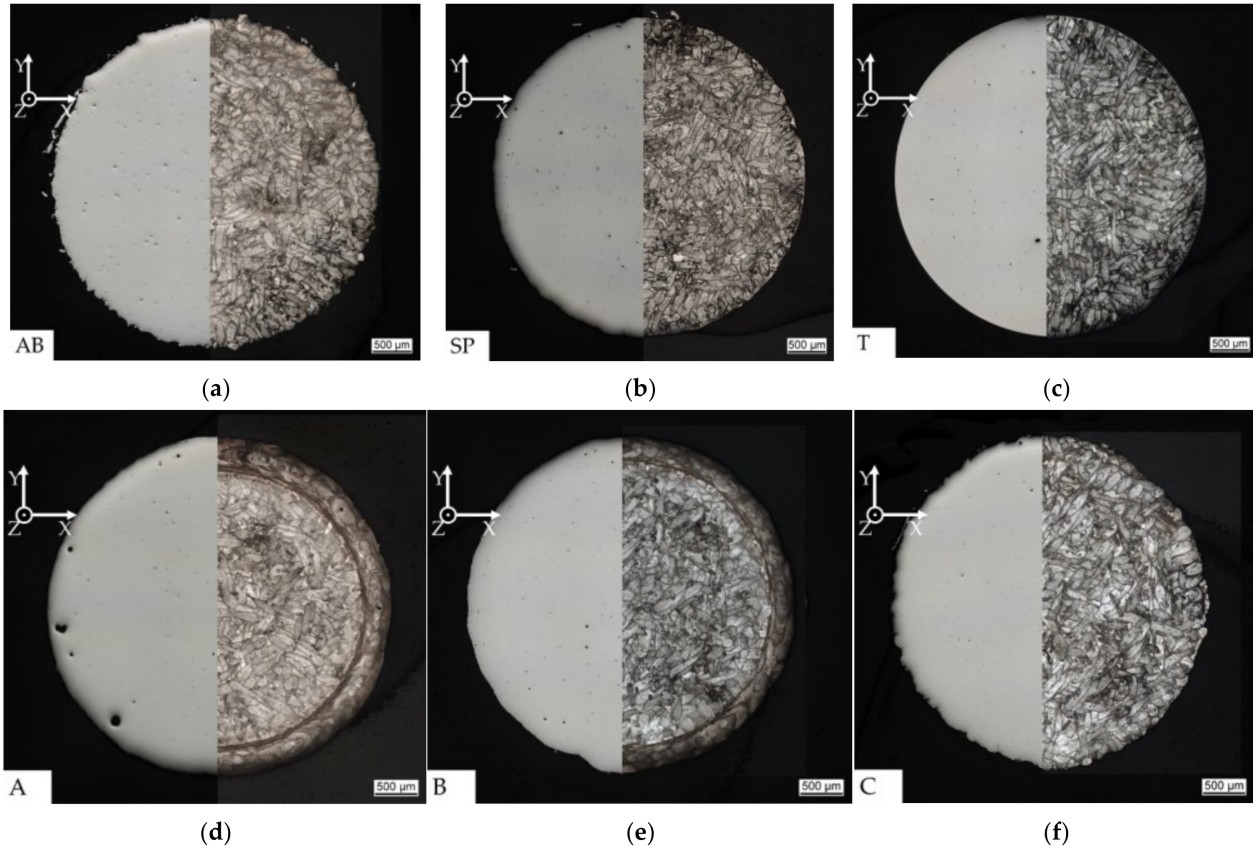

**Figure 5.** Overview of the metallographic cross sections of the (**a**) as-built-surface AB for comparison and the two post-process-surface states (**b**) shot peened SP, (**c**) turned T, (**d**) contour parameter A, (**e**) contour parameter B, (**f**) contour parameter C.

In Figure 6 the distribution of the pores evaluated on the reconstructed CT-scans of the specimens in dependence of the distance of the surface, is illustrated. It can be stated that on specimens built with contour parameters A and B, the majority of the pores and the pores with larger volume are located in the area near the surface, which is caused by the contour scan. Only on specimens with contour parameter set C are the pores evenly distributed throughout the specimen. This indicates that the line energy of the contour parameter set A is high enough that keyhole-welding-mode described in [43,44] occurs and forms around pores with larger volumes. Although the near-surface porosity is significantly decreased with contour parameter set B, the amount of porosity still indicates that the reduction of the line energy was not sufficient to prevent pores. However, on contour parameter set C, the contour line energy was so low that the infill parameter set re-melted the contour scan, which prevents keyhole-pore formation, but results in a significantly higher surface roughness due to the adhesion of not fully melted powder to the surface.

### 3.3. Residual Stress Analysis

In Figure 7a, residual stress–depth profiles in the measuring area of the AB- and post-processed (turning and shot-peening) fatigue specimens can be observed. It is noticeable that the residual stresses of the AB-specimen change from low tensile residual stresses at the surface to nearly no residual stresses with increasing depth. This behaviour was also observed and simulated in [57] and can be explained by the higher cooling rate at the surface: Due to the higher thermal conductivity of the material than the ambient air, inner areas of the specimens cool down slower, because the heat flows through the centre of the specimen into the building platform, which results in higher residual stresses at the surface due to more rapid cooling than the centre. The material has no time to assume the most

energetically favourable status, causing residual stresses to build up on the surface. The residual stress–depth curve of the turned specimen shows a small area of residual stresses at the surface, probably due to the induced material deformation owing to the turning process (a). The shot peened specimen shows high compressive stresses at the surface. The zero crossing of the residual stresses curve was shifted approximately 400 μm into the subsurface area.

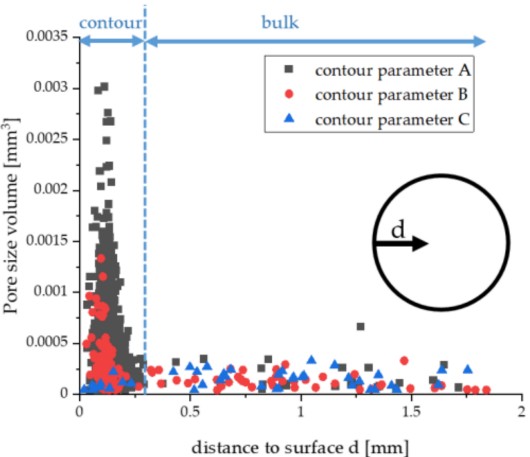

**Figure 6.** Pore size distribution of the manufactured specimens with differing contour parameters dependent of the distance to the surface.

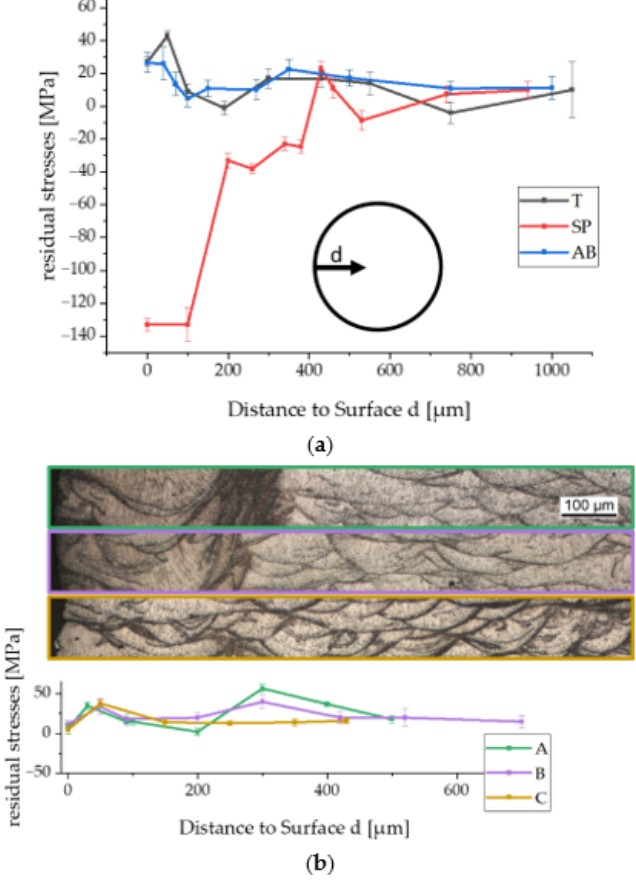

**Figure 7.** (**a**) Residual stress–depth profiles of as-built (AB), shot peened (SP) and turned (T) specimens. (**b**) Residual stress–depth profile of contour parameters A, B and C. The curves call the gradient.

In Figure 7b, the residual stress–depth profile of specimens with contour parameters A, B, and C, and the surface area of the etched square sections, is shown. Contour parameter A shows a first maximum at approximately 40 μm near the surface, similar to the AB-specimens, which can be explained by the effect described above for the AB-specimens. At approximately 350 μm depth, a second maximum of approximately 60 MPa tensile residual stresses were identified, which is most likely induced by the post-contour parameters: The re-melting of the material results in a locally rapid cooling at the interface of the contour to the bulk, which results in high residual stresses. The re-melted surface area can be seen in the etched cross section above the graph in Figure 7b, green. The heat treatment after the manufacturing process was not sufficient to remove those residual stresses. Specimens built with contour parameter set B show the same second peak at approx. 350 μm, but the induced residual stresses are significantly lower in comparison to contour parameter A. This leads to the assumption that a lower line energy induces fewer residual stresses. Likewise, the contour on the B-specimens in the cross section in the etched state is visible (Figure 7b, purple). Specimens built with contour parameter C do not show the second peak. In comparison to A and B, only a pre-contour-parameter was used. The area of the contour was completely remelted, which is the reason why no contour can be made visible in the square section of specimens with contour parameter set C (Figure 7b, yellow) in comparison to A and B.

### 3.4. Fatigue Investigations

The fatigue results of the specimens with differing surface roughness due to turning (T) and shot peening (SP) in comparison to the as-built state without contour (AB) are summarized in Figure 8. It is evident that all fatigue tests reveal a high scatter, even the turned specimens. The fatigue strength limit for AB is around 40 MPa with a steep time strength area in comparison to shot peened and turned specimens. The fatigue strength area was more than doubled with shot peening SP or turning T, which is a similar result in comparison to AlSi10Mg [40,42]. The fatigue strength of SP exhibits the highest fatigue strength of all test series in this work and even exceeds the fatigue strength of turned specimens. This behaviour indicates a significant influence of surface roughness, residual stresses, and porosity on fatigue behaviour.

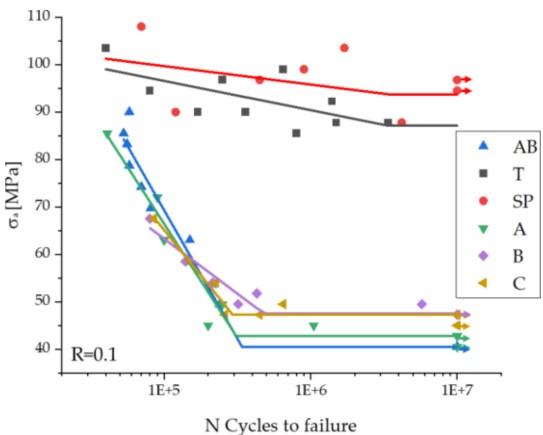

**Figure 8.** Results of the fatigue tests with R = 0.1 for the post-process treatments turning (T) and shot peening (SP) in comparison to specimens with as-built surface (AB) gathered with the bead string method acc. to DIN 50100 [53]. Run-throughs are marked with an arrow. Fatigue results of the in-process treatments of contour parameter A, B, and C in comparison to turned specimens (T) and as-built surface specimens (AB). The lines are only for visualisation.

The as-built specimens without contour parameter sets (AB) have the highest surface roughness of all tested specimens, which results in the lowest fatigue strength. In Figure 9a the exemplary fracture surface of a specimen without contour parameters was investigated

by SEM. A fatigue fracture with ductile residual force fracture according to VDI 3822 Part 2 [56] can be obtained. Fracture origins were located by fracture paths running towards them, which are caused by the crack propagation. Multiple fracture origins (exemplary view in Figure 9b) are located at the circumference of the surface, which indicates that the notches due to the high roughness promote the crack initiation, which was stated by Murakami [34] as a characteristic property of AM-materials in general. The multiple cracks then unite and reduce the surface area, which overall results in a poor fatigue strength.

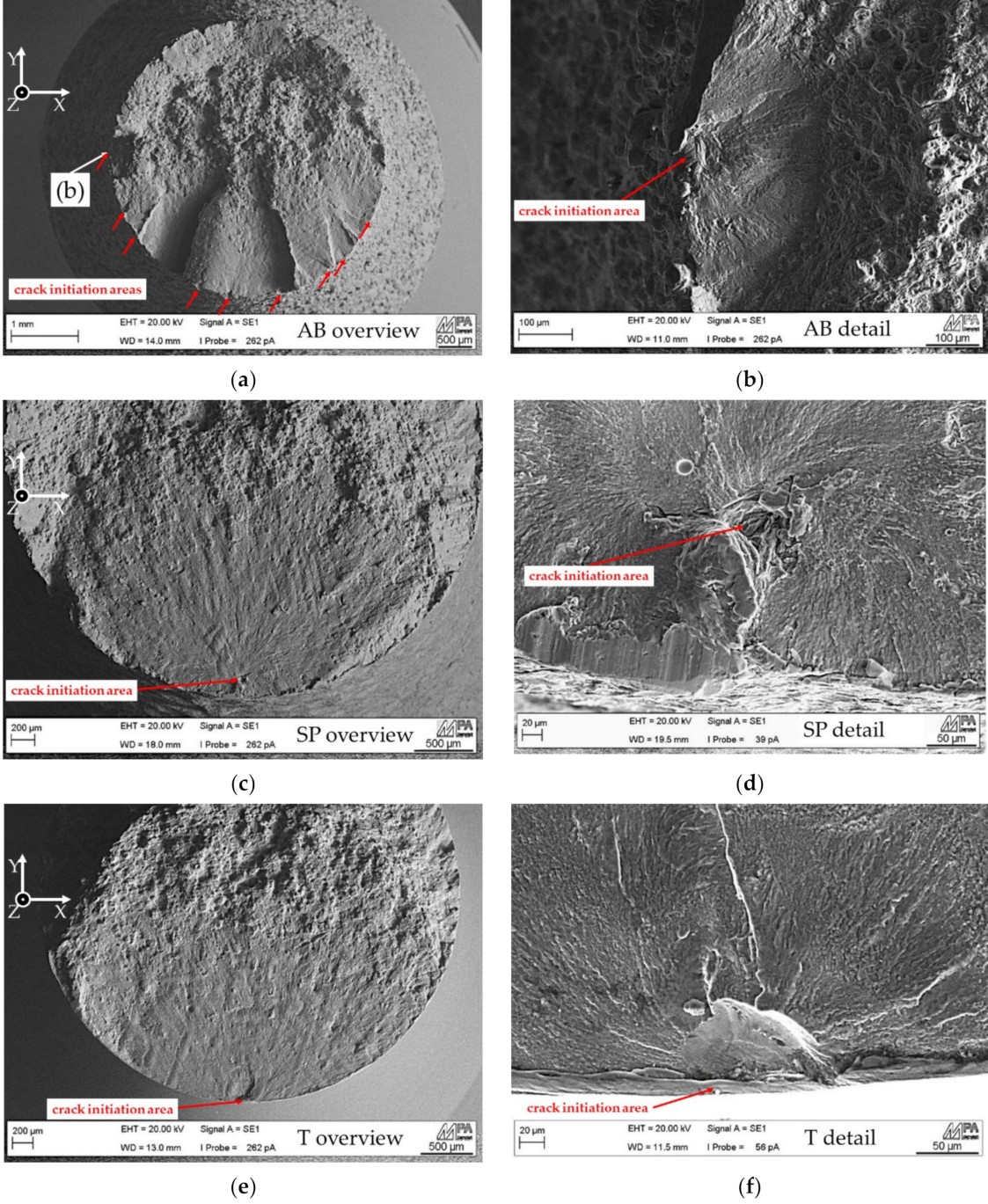

**Figure 9.** (**a**) Exemplary overview of a fracture surface of an as-built specimen AB. (**b**) AB detailed view of a crack initiation area, (**c**) exemplary overview of a fracture surface of a shot peened specimen SP, (**d**) SP detailed view of the crack initiation area, (**e**) exemplary overview of a fracture surface of a turned specimen T, (**f**) T detailed view of a crack initiation area. All fracture surfaces inside one charge are of similar character, regardless of the stress amplitude. R = 0.1.

With shot peening, the surface with lower roughness and the compressive residual stresses are improving the fatigue properties significantly (SP in Figure 8), which is a similar behaviour as found by AlSi10Mg [42]. In Figure 9c, an exemplary overview of an SP fracture surface investigated by SEM can be seen. Unlike AB, only one specific fracture origin on every tested specimen is present, which can exemplarily be seen in detail in Figure 9d. It is noticeable that the fracture origin is located in all investigated specimens approximately 100 μm below the surface. This indicates that the shot peening increases the resistance against crack initiation due to the induction of compressive residual stresses. Pores or oxides in the subsurface area, where the compressive stresses are lower or even in the tensile range, then act as crack initiation areas (residual stresses in Figure 7 and fracture surface in Figure 9c). The ruptured surface area, which can be seen in Figure 10e, does not exhibit microscopic or macroscopic notches in comparison to AB. These findings agree with [58], who showed the same effect on LPBF-IN718, where the shot peening shifts the positive residual stresses from the surface towards the inside of the specimens. In the case of IN718, this effect even lowers the fatigue properties due to higher notch sensitivity.

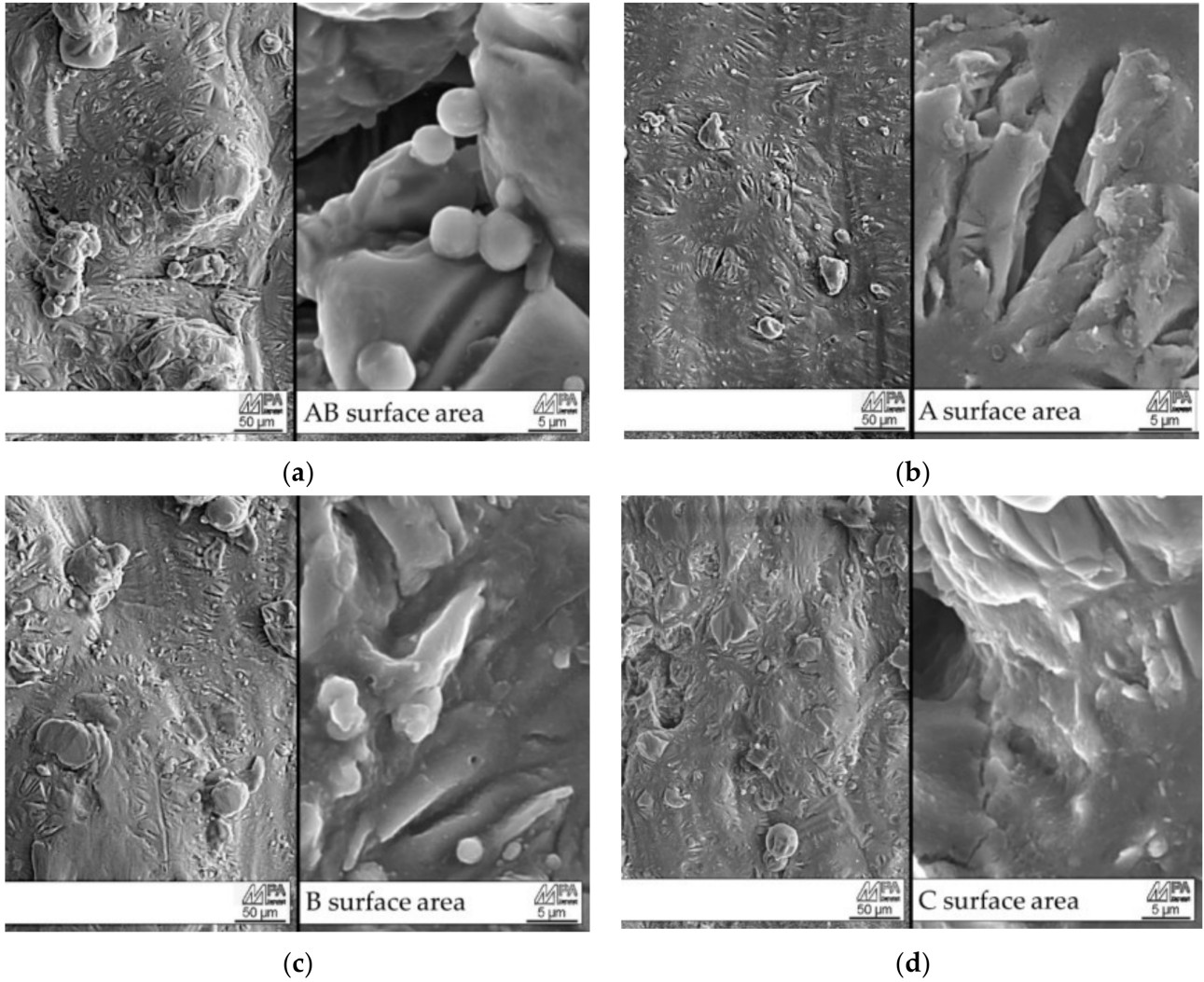

(a)

(b)

(c)

(d)

**Figure 10.** *Cont.*

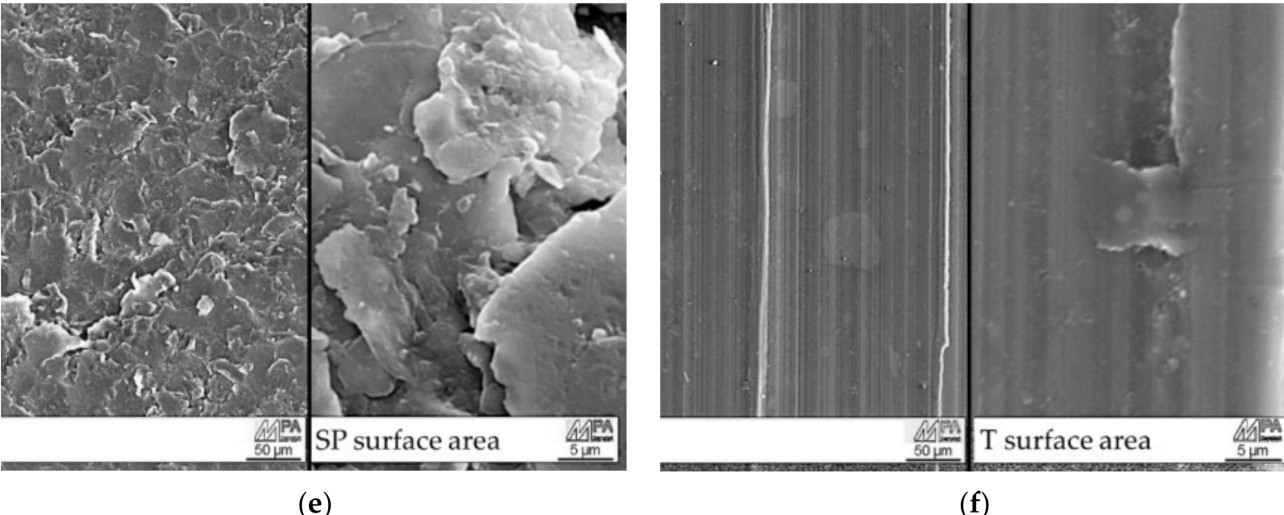

(**e**)                (**f**)

**Figure 10.** Exemplary SEM-images of the surfaces before the fatigue test of (**a**) as-built (AB), contour parameter sets (**b**) A, (**c**) B, (**d**) C, post process improvements, (**e**) shot peened (SP), and (**f**) turned (T) specimens.

Turned specimens exhibit the lowest surface roughness of all tested specimens. Only the slight grooves of the machining induce recurring peaks and valleys together with the inner porosity or imperfections that were converted to open porosity at the surface due to the machining process. The resulting surface area can be seen in Figure 10f, showing an exemplary imperfection (Figure 10f, right hand side). In comparison to shot peened specimens, the overall surface roughness is lower (Table 6), but a certain amount of tensile residual stresses at the surface was induced due to the turning process (Figure 7). In Figure 9e, an exemplary SEM-image of the fracture surface of a turned specimen can be seen. Even at lower stress amplitudes, the crack initiation area is always located at the surface. The cracks initiate on small imperfections at the surface, such as little breakouts or open porosity, which can exemplarily be seen in Figure 9f. Aboulkhair et al. [41] investigated the same crack initiation areas on turned fatigue specimens for material AlSi10Mg. The open porosity and imperfections at the surface act as notches and are therefore the most likely crack initiation areas. Additionally, the induced residual stresses due to the machining process increases the probability that the crack initiates at surface defects even further.

In comparison to the post-process parameters tested in this work, the in-process contour parameters only increase fatigue strength by about 5% to 20%.

Contour parameter A has a similar time strength area as AB and an only 3 MPa higher fatigue strength. B and C show slightly higher fatigue strengths. Specimens with contour parameters B and C have the same fatigue strength of approximately 48 MPa, but the time strength graph of C is steeper.

It is evident that the developed contour parameters only slightly increase the fatigue strength in comparison to those of as-built specimens without a contour of about 3 MPa to 8 MPa. Despite the significant reduction of surface roughness with contour parameter A in comparison to AB, the fatigue strength is only increased by 3 MPa. Nevertheless, the remaining notches at the surface are sufficient to act as crack initiation areas; in Figure 10a, SEM-images of the surface of a specimen AB can be obtained. It can be investigated that the high roughness and powder agglomerations (spherical material) lead to large notches. On the surface of specimen A (Figure 10b), not as many powder agglomerations or as much roughness can be investigated, therefore it can be seen that the surface is littered with microscopic notches, which have a high notch factor, which lowers the fatigue strength of the material [59].

In Figure 11a,b, an exemplary fracture surface of A can be seen. The subsurface porosity present in this specimen, resulting from the contour scan with high line energy (Figure 5d,e and Figure 6), does not act as crack initiation areas. This leads to the assumption

that sharp, microscopic notches together with higher tensile residual stresses at the surface, in comparison to the subsurface area (Figure 7), have a significantly higher impact on fatigue strength than subsurface pores at R = 0.1.

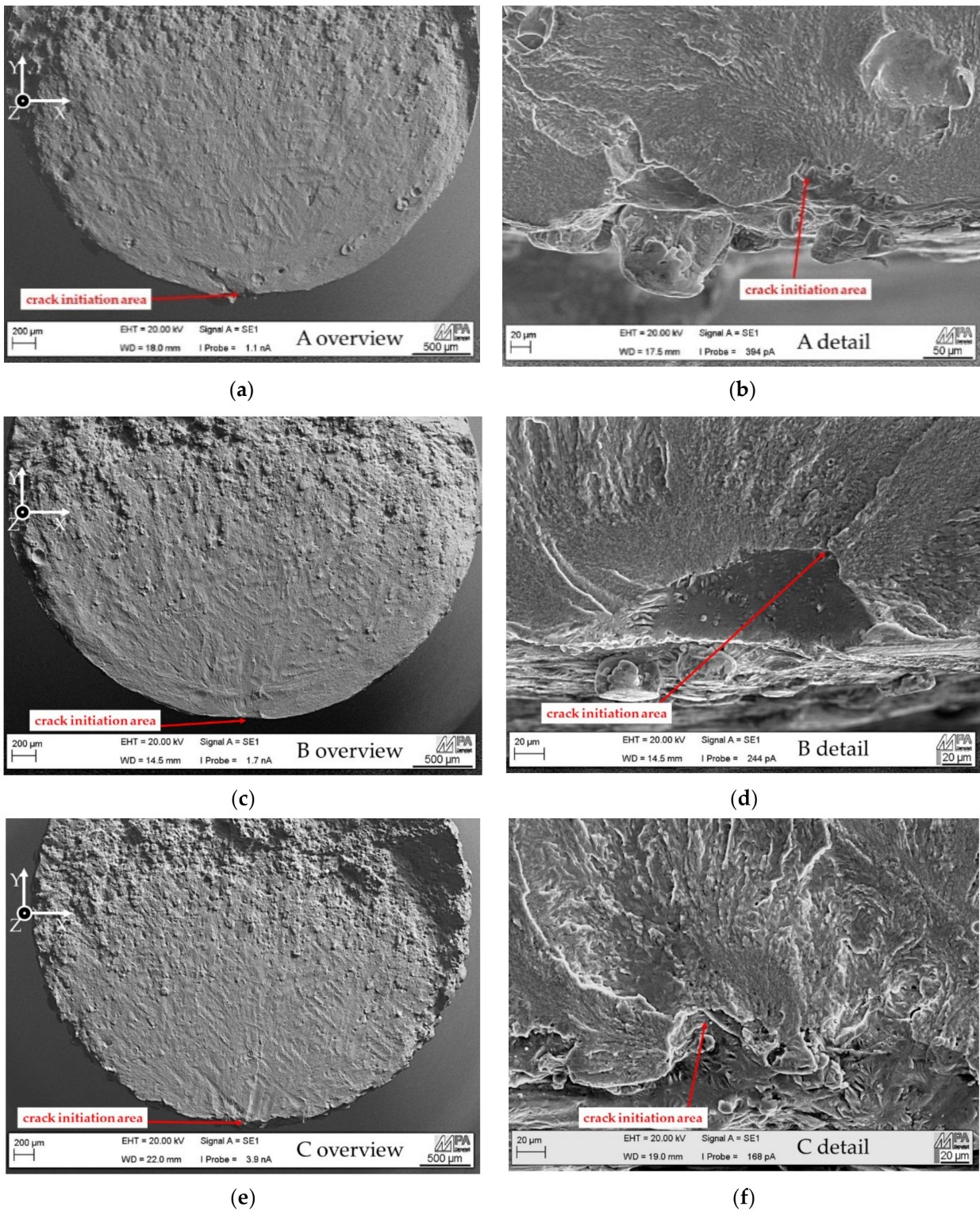

**Figure 11.** (**a**) Exemplary overview of a breaking surface of A. (**b**) Detailed view of a crack initiation area of A, (**c**) exemplary overview of a breaking surface of B, (**d**) detailed view of the crack initiation area of B, (**e**) exemplary overview of a breaking surface of C, (**f**) crack initiation area of C. All breaking surfaces inside one charge are of similar character, regardless of the stress amplitude. R = 0.1.

In Figure 11c, an exemplary overview of the fracture surface of B can be seen. Similar to turned specimens in the detail view in Figure 11d, it can be gathered that crack initiation was at an open pore at the surface of the specimen, which indicates that the high porosity at the surface and subsurface in comparison to C (Figure 6) lowers the fatigue strength. In the exemplary overview (Figure 11e) and detailed view (Figure 11f) of the fracture surface of C, it can be obtained that the crack initiation area was at a notch due to the high roughness.

The fatigue strength of B and C is nearly similar, because the lower surface roughness on B is compensated by its higher subsurface porosity (Figure 6) of B in comparison to C: Either the rough surface or porosity in the surface area act as notches and therefore as crack initiation areas. Regarding the microscopic investigations on the surface area of specimens B (Figure 10c) and C (Figure 10d) in comparison to A or AB, no microscopic notches and no high amount of powder agglomerations can be found. The fatigue strength is not increased in the same manner as it is for post-process parameters such as turning or shot peening. In the case of turning, the porosity and roughness of the surface is nearly removed and in the case of shot peening the compressive residual stresses increase the crack initiation resistance.

## 4. Conclusions

In this work, specimens of Scalmalloy® with differing surface states were manufactured by LPBF in order to investigate the influence of the surface condition and the influence of in-process optimization strategies and post-process improvements of the surface on the fatigue behaviour. Therefore, three different contour parameters and a turning and shot peening process were compared. Metallographic analysis, fracture surface analysis by SEM, residual stress measurements via XRD, roughness measurements, porosity analysis via CT, and tensile and fatigue tests were performed. The following findings were achieved:

- Compared to the as-built surface state that only had infill parameters, the post processing steps using shot peening and turning increases the fatigue strength by a factor >2. In contrast, the use of contour parameters improves fatigue properties by only around 10 to 20% in comparison to the as-built state.
- Crack initiation on contour-parameter specimens occur on sharp, microscopic notches at the surface together with high tensile residual stresses. Even subsurface pores induced by a contour scan in keyhole-welding-mode do not change the crack initiation area at the surface.
- A contour scan with low line energy results in the same fatigue strength as a contour scan with medium line energy, despite the higher roughness, because the overall porosity, even in the surface area, is lowest, which results in less supporting crack initiation areas in combination with the sharp notches.
- Shot peened and turned specimens show comparable fatigue behaviours, despite the higher roughness of shot peened specimens: This disadvantage is compensated by the induced compressive residual stresses at the surface, which lowers the tendency for crack initiation at the surface and shifts the crack initiation area into the specimen.

This study shows that a significant improvement of the fatigue strength can only be achieved due to conventional post-processing surface treatments. The best results for improving the fatigue strength with in-process parameters could be achieved with a contour scan with low and medium line energies that are not in the keyhole-welding-mode. It is important to note that only fatigue tests under a tension swell load of $R = 0.1$ were performed. Further investigations have to be conducted in order to reveal whether tension–compression load leads to identical results. The subject of the current research is to investigate whether a poor contour scan along with shot peening might be counterproductive, because the tensile residual stresses might be superimposed by the local stresses at subsurface porosity.

**Author Contributions:** Conceptualization, J.M., T.R., and H.C.H.; investigation, J.M. and T.R.; writing—original draft preparation, J.M.; writing—review and editing, T.R., H.C.H., M.O., M.W., and E.A.; funding acquisition, H.C.H.; supervision, M.O. All authors have read and agreed to the published version of the manuscript.

**Funding:** The research and development project 'BadgeB'that forms the basis for this publication is funded within the scope of the "Additive Fertigung—Individualisierte Produkte, komplexe Massen-produkte, innovatiove Materialien" (Pro Mat 3D) by the Federal Ministry of Education and Research, funding no. 02P15B154. The project BadgeB is managed by the KIT project management agency "Projektträger Karlsruhe—Produktion und Fertigungstechnologien". The authors are responsible for the content of this publication.

**Data Availability Statement:** The data presented in this study are available on request from the corresponding author.

**Conflicts of Interest:** The funders had no role in the design of the study; in the collection, analyses, or interpretation of data; in the writing of the manuscript, or in the decision to publish the results. The authors declare no conflict of interest.

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
