# Peer review of "Influence of LPBF-Surface Characteristics on Fatigue Properties of Scalmalloy®"

_metals, doi:10.3390/met11121961_

Round 1

Reviewer 1 Report

In the manuscript «Influence of LPBF-Surface Characteristics on Fatigue Properties of Scalmalloy», the authors investigated the effect of surface treatment on the final fatigue properties of manufactured LPBF specimens. Currently, the production of parts by selective laser melting is an urgent topic for the development of industry. It should be noted that in addition to using various methods of surface treatment, the authors studied in detail the parameters of the obtained samples, such as roughness, microstructure, distribution of residual stresses, fatigue properties, and so on. All the results obtained have been thoroughly studied by the authors and discussed. The conclusions are confirmed by the obtained results.

I have one small comment on the manuscript:

The text contains many inaccuracies in the format. Authors should carefully check the text. For example: line 149 – “Error! Reference source not found..”, Line 170,172 – “ (… Figure 1).

Author Response

Dear Editor,

thank You for reviewing the article.

Unfortunately I could not find any corrupted links on my manuscript. Maybe the versions of Word or Acrobat are different and causing issues.

To address this issue, I erased all Content Control Elements in Word and I hope this will solve the problem.

Additionally I added some page breaks for better readability.

Best regards,

Jens Musekamp

Reviewer 2 Report

This manuscript conducts a detailed examination on the effect of turning and shot peening parameters on fatigue properties of additively manufactured alloys. An in depth metallographic analysis is conducted with extensive experimental efforts. This work shows the significance of post surface treatment process and is clearly is of interest to a wide range of readers. The manuscript is in scope with the journal. Therefore, the manuscript is recommended for acceptance provided the following minor comments are addressed:

  • The manuscript is poorly formatted, which makes it very hard for readers to concentrate on the contents of the work. For example, problems shown in lines 149, 170, 209, 345, 357, and so on are not the best way to present a manuscript for peer-review.
  • Section “3.2 Residual stress Analysis” should be “3.3 Residual stress analysis”
  • It seems like a new section should be introduced in page 13 since the paragraphs seem to describe the results on fatigue strength.
  • The lines in figure 7(a) and (b) seem to be fitted with respect to the data points. It is recommended to use linear line segments between data without smooth fitting.

Author Response

Dear Editor,

thank You for reviewing the article.

The manuscript was formatted and now it should be in a better way to present it. Also the Section 3.2 was changed into 3.3. The new section „3.4. Fatigue Investigations“ was added again on page 13, which somehow got lost while formatting. Thank You for this hint.

The lines in figure 7a and 7b are now fitted with respect to the data points.

Best regards,

Jens Musekamp

Reviewer 3 Report

This manuscript presents the effects of surface roughness, residual stress and porosity on the fatigue properties of Al-Mg-Sc-alloy Scalmalloy. The properties are examined using roughness measurements, residual stress measurements, optical Microscopy (OM), fracture surface analyses by means of Scanning Electron Microscopy (SEM), tensile tests and fatigue tests. This study has certain significance for the optimization of the fatigue strength of the alloy. It is a topic of interest to the researchers in the related areas but the paper needs minor improvement before acceptance for publication. My detailed comments are as follows:

1. In page 8, the author mentioned that "the dimple is judged as transgranular fracture behavior according to the dimple characteristics". What are the experimental results or references supporting the author's conclusion? Please add a local enlarged figure to further analyze the intergranular fracture mechanism.

2. In Figure. 8, please explain the fitting method of scatter diagram curve. The curve shall be further fitted by high-order polynomial to better reflect the relationship between fatigue loading times and stress state.

3. Figure. 9 shows the effect of roughness on fracture by judging the location of fatigue crack source and the aggregation mode of surface microcracks. The evidence is not sufficient. Please make further supplement.

4. The fracture shown in Figure.11 is analyzed in page 16, indicating the influence of surface roughness on fracture. The spherical material in the figure is not described, and its influence on fracture needs to be further described.

5. In this paper, the effects of surface roughness, residual stress and porosity on the fatigue properties of the alloy are discussed. But the accurate and better process parameters are not given in the conclusion. Please comprehensively consider the influence of various parameters and give clear parameters to optimize the fatigue performance of the alloy.

Author Response

Dear Editor,

thank You for reviewing the article and the constructive comments. The comments were included as follows:

To point 1:

The reference supporting the conclusion was added.

To point 2:

The execution of the experiments were performed according to the bead string method and was evaluated according to DIN 50100. This procedure is described in „Materials and Methods“ in line 259/260. For better clarity the reference was linked in the description of the figure.

To point 3:

In VDI 3822 Part 2 typical fracture surfaces of fatigue fracture with ductile residual force fracture like in the fatigue specimens in this work are displayed (page 30). It can also be obtained that the specimens show a circumferential bending fatigue fracture surface. Fracture paths pointing to the fracture origin (local notches or material inhomogeneities). A sentence for the explanation with reference was added in Line 401 to 403.

To point 4:

The spherical material are powder agglomerations at the surface of the specimens which lead to notches at the surface. Due to the fact that the notches are round, the notch factor is not as high as on sharper notches like Figure 11b, right. Nevertheless, they could also have an impact on the fatigue strength due to the fact that they form large notches. This instance was described in Line 460. Some sentences were added to further describe the influence in line 463-464.

To point 5:

The best results regarding the fatigue strength show the post process surface treatments, which was already stated in Line 517 to 518 in the conclusion. The best in-process-parameters are contour scans with medium to low line energy that are not in keyhole-welding-mode. A sentence in addition was added in Line 518 to 520.

Best regards,

Jens Musekamp